# An Examination of Child Pedestrian Rule Compliance at Crosswalks around Parks in Montreal, Canada

**DOI:** 10.3390/ijerph192113784

**Published:** 2022-10-23

**Authors:** Marie-Soleil Cloutier, Mojgan Rafiei, Lambert Desrosiers-Gaudette, Zeinab AliYas

**Affiliations:** 1Institut National de la Recherche Scientifique, Centre Urbanisation Culture Société, Montréal, QC H2X 1E3, Canada; 2Centre de Recherche en Santé Publique (CReSP), Université de Montréal, Montréal, QC H3N 1X9, Canada

**Keywords:** road traffic safety, crossing behavior, child pedestrian, rule compliance, crosswalk, parks

## Abstract

This study aims to examine child pedestrian safety around parks by considering four rule-compliance measures: temporal, spatial, velocity and visual search compliance. In this regard, street crossing observations of 731 children were recorded at 17 crosswalks around four parks in Montreal, Canada. Information on child behaviors, road features, and pedestrian–vehicle interactions were gathered in three separate forms. Chi-square tests were used to highlight the individual, situational, behavioral and road environmental characteristics that are associated with pedestrian rule compliance. About half of our sampled children started crossing at the same time as the adults who accompanied them, but more rule violations were observed when the adult initiated the crossing. The child’s gender did not have a significant impact on rule compliance. Several variables were positively associated with rule compliance: stopping at the curb before crossing, close parental supervision, and pedestrian countdown signals. Pedestrian–car interaction had a mixed impact on rule compliance. Overall, rule compliance among children was high for each of our indicators, but about two-thirds failed to comply with all four indicators. A few measures, such as longer crossing signals and pedestrian countdown displays at traffic lights, may help to increase rule compliance and, ultimately, provide safer access to parks.

## 1. Introduction

In Canada, traffic collisions are the leading cause of injury-related death for children under 14 [1]. On average, 30 child pedestrians are killed and more than 2000 are injured every year, as Canada lags behind the OECD’s top performers for the past years [2]. A great proportion of these collisions occur at road intersections [3], and in a highly thorough study on child traffic safety published in 2021, pedestrian crosswalks were found to be particularly dangerous locations for children to be injured [4].

Crossing a street involves a complex series of tasks—i.e.: detecting traffic, planning one’s route, assessing speed and traffic, making oneself visible—that exacerbates the risk of injury for children [5]. Hence, because of their small stature and their developing overall physical and cognitive attributes, child pedestrians form a vulnerable road-user group at risk of severe injuries with long-term physical and mental impairments [6,7].

Road insecurity while crossing streets is a well-founded reason for children to avoid walking or for parents to drive children instead of having them walk to different destinations like schools or parks [8]. Among the most frequent destinations for children, scientific literature has focused extensively on road safety near schools (see for example the systematic review by Rothman et al. [9]). However, much less attention has been given to parks despite the fact that many children frequent them after school or on weekends—especially in dense urban areas where there are no yards for playing [10,11]. In addition, the number of park visitation is affected by having safe and accessible routes for pedestrians [12,13] and since bicycle and pedestrian accidents are more frequent around parks than they are elsewhere in cities, it has been recognised that roadway safety around parks needs special attention [8,14]. Accordingly, a study published in 2017 found that the risk of child pedestrian fatalities is greater around parks: 1.04 to 2.23 times higher than around schools and 1.16 to 1.81 times higher than any other city-wide crossing [8], reiterating the pressing need to study road safety around parks.

For children, injury prevention is often based on systematic behavioral rule adherence [15]. Low levels of compliance with road rules and unsafe behaviors from either drivers or pedestrians are the main reasons for low pedestrian safety levels [16]. In other words, when pedestrian and motor vehicle users comply with crosswalk rules, pedestrian safety increases [17]. Accordingly, a few studies address the prevalence of traffic violations among pedestrians based on specific individual characteristics such as age or gender [18,19]. Since there is little research on compliance to rules during childhood, our understanding of how various pedestrian and road environment characteristics affect a child’s compliance to road safety rules is rather limited. The current study attempts to fill this gap regarding child pedestrian safety around parks by examining individual, situational, behavioral and road environment characteristics that determine compliance with various road safety rules during street crossings.

## 2. Factors Associated with Child Pedestrian Safety and Compliance

Past research on child pedestrian injuries demonstrates that risk factors fit into one of four categories, and that these have remained unchanged for decades—road accidents involving children are caused by a combination of individual, situational, behavioral and physical (road) environment characteristics. Major risk factors related to rule compliance for each of these categories are presented. We acknowledge that there are more risk factors not mentioned here (for a broader portrait, see for example: [9,20,21]).

### 2.1. Individual Characteristics

Demographic characteristics such as age and gender are recognized as important predictors of child pedestrian injuries [22]. Several studies attribute increased road injury risk experienced by younger pedestrian children to their lack of traffic knowledge and experience, cognitive and physical ability, and visual acuity [4,23,24] According to Tabibi and Pfeffer’s [25] analysis of the effect of children’s ages on their ability to distinguish between safe and risky road crossing places, the outcomes indicated that the ability improves starting around the age of 10 or 11. Several studies came to the similar conclusions that there are no gender differences in the capacity to judge whether a road-crossing site is safe or harmful for children [26,27]. However, some concluded that boys and girls of various ages showed varied behaviors near and on roadways [28,29]. For example, Barton and Schwebel [30] and Granié [31] found that boy pedestrians are less likely to comply with road safety rules and more likely to be involved in injury-related accidents.

### 2.2. Situational Characteristics

Situational conditions can influence safety and compliance [32]. It has been documented that children are less adept than adults in evaluating the risks associated with the road environment as they have weak visual search strategies and are less able to spot dangerous circumstances [33,34], making it important to have supervision, at least until children have more experience. When adults accompany children to and from their destinations, there is a demonstrated reduction in the risk of injury [30,35]. We hypothesize that the parent/caregiver’s gender may also have an impact on rule compliance, as men display a more careless attitude and are more frequently guilty of violations [36,37]. Likewise, the presence of other pedestrians also crossing may influence crossing speed, timing, trajectory, and level of attention [38].

The term “interaction” usually refers to an event where, without any collision, the paths of both a vehicle and a pedestrian intersect while they are still on the roadway [39]. As interactions are correlated to more collisions [40,41], the occurrence of such interactions may alter pedestrian behavior which may, in turn, lead to more collisions [42,43].

### 2.3. Behavioral Characteristics

Speed (i.e.,: tempo) before and after crossing (walking or running), failure to stop at the curb, failure to look before crossing, and attempting to cross when a car is near are considered unsafe behaviors since they reduce the ability to correctly assess traffic situations [15,44]. Given that behavior and judgement are inherently inconsistent in young age groups, child pedestrians are most notably at risk. Crossing in a straight line (not diagonally) and waiting for the next green light at the curb are known to be related to fewer interactions with vehicles and therefore reduce the risk of collision [45].

### 2.4. Road Environment Characteristics

The pedestrian crosswalk, its design (length and width), the method of control at the crosswalk (traffic signals, stop sign), and the presence of countdown timers are all significant factors in how children behave as pedestrians and, consequently, how safe they are in pedestrian crossing zones. Accordingly, uncontrolled crosswalks inflate the risk of conflict, especially in urban areas [46]. Crosswalk width also impacts safety since wider streets expose pedestrians to traffic for longer [47] and pedestrians tend to cross such streets faster and more carelessly, leading to possible dangerous behavior [48].

Pedestrian signals seem to positively affect safety as pedestrians are less likely to finish crossing on a red light when such signals are present [49]. The results from countdown timers, however, are highly contradictory. Although they demonstrate an increase in safe behaviors [49,50,51]—including child pedestrians who tend to finish crossing on time where these timers are present [52]—they also give rise to non-compliant behaviors [53,54], and lead to an increase in the number of late-starter and late-finisher pedestrians [55]. Finally, the walking speed generally used to calculate the time required to cross fully at a light-controlled intersection is 1.2 m per second [56]. This speed does not take slower walkers or various contextual characteristics into consideration. Walking speed varies by age (children being slower), group size and composition, traffic-control conditions and even departure signals [26,57]. Such characteristics can affect the number of collisions and injuries.

## 3. Methods

### 3.1. Site Selection

The study territory consists of two central boroughs on the Island of Montreal in Canada: Villeray/Saint-Michel/Parc Extension and Rosemont/Petite-Patrie (Figure 1). Visited parks were selected using a two-step process. First, we randomly selected four types of parks in both neighborhoods based on the typology developed by Apparicio et al. [58]: Group A (very small parks with a playground, n = 5), B (small parks with 2 facilities: playground and sports field, n = 5), C (small parks with 3 facilities, n = 4) and D (small parks with 2 facilities, including an ice rink or pool, n = 4). One park was chosen in each category (n = 4 total: see Figure 1). Second, we visited each of the selected parks to explore crosswalk features and child pedestrian presence. In this regard, adjacent intersections and crosswalks (n = 17) were selected to represent a variety of crosswalk signage, road types and distances to the entrance of the park when observing children walking.

### 3.2. Observation Protocol

Observations of child pedestrians crossing towards the park were recorded between June and August 2017, during the daytime, on weekdays and weekends (between 9 a.m. and 7 p.m., minimum 3 h per day of observation and 2 days per site, minimum 40 children at each site). Four trained observers were posted near the sidewalk or in the park toward which the child pedestrian was heading and would work in groups of two at busy intersections. If there was more than one child or a group of children, only one of them was randomly selected for observation. All observers were trained at the same time for all of the observed items, including child age category. All items in the observation forms were validated onsite until all of the observers had the same answers (3 periods of 3 h). All observations were recorded on iPads using the Survey123 software program developed by the ESRI [59]. Each child and each crosswalk were given a unique ID, which permit to merge together the three data sources (see below the description) after the data collection.

Crossing situations were recorded using three different tools from previous research [42]:

*Child pedestrian crossing behaviors* were observed at three specific times (Figure 2): at the curb, on the crosswalk, after crossing. Other individual and situational characteristics were recorded for each observed children.Four *crosswalk characteristics* (Table 1) were recorded and used for our analysis: presence and type of traffic control sign (stop sign, traffic light, pedestrian light), crosswalk width (in meters), time permitted to cross (in seconds), and distance between the nearest entrance of the park and the crossing (see Figure 3 for examples). For street crossings with a traffic light, it was possible to calculate the ‘required speed to cross in time’: by dividing the crosswalk width by the time permitted to cross (pedestrian or green phase).*Interactions between the child pedestrian and vehicles* was recorded when the pedestrian’s path and the driver’s path crossed while the pedestrian was still on the street (on the pavement, not curb). This broader definition of conflicts was used to be able to capture events with “dangerous proximity” [60], which can be considered serious incidents, particularly for child pedestrians.

**Figure 2 ijerph-19-13784-f002:**
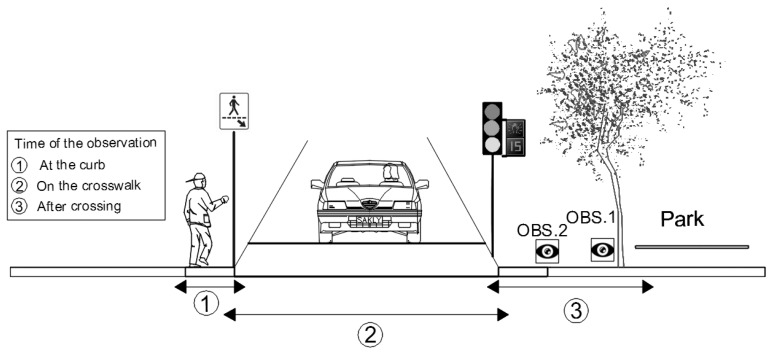
Observation protocol related to child pedestrian crossings.

**Figure 3 ijerph-19-13784-f003:**
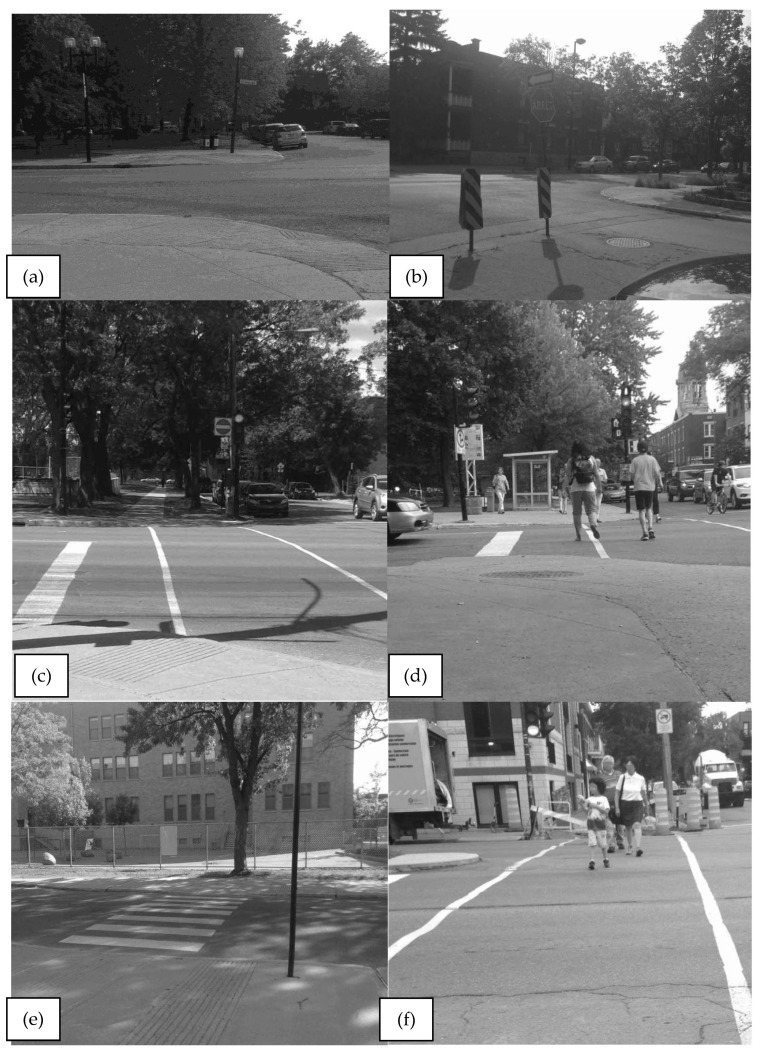
Example of crosswalk characteristics: (**a**) No signage (**b**) Stop sign (**c**) Traffic light without pedestrian light (**d**) Traffic light with pedestrian countdown display (**e**) Narrow crosswalk (**f**) Wider crosswalk.

**Table 1 ijerph-19-13784-t001:** Crossings characteristics and number of crosswalks.

Characteristics	Number of Crosswalks
Signage	
No signage	2
Stop sign	2
Traffic light without pedestrian light	5
Traffic light with pedestrian countdown display	8
Crosswalk width	
Less than 15 m	6
Between 15 and 25 m	9
More than 25 m	2
Required speed to cross in time (traffic lights only)	
1 m/s or less	9
More than 1 m/s	4
Distance between nearest entrance and intersection	
5 m or less	15
More than 5 m	2

### 3.3. Rule Compliance

To account for child adherence to pedestrian safety rules, we created four binary composite indicators that distinguished child pedestrians based on temporal compliance, spatial compliance, velocity compliance, and visual search. The four rule compliance indicators are original constructs based on the combination of the literature review on children’s risky behaviour and data availability within our observation tool. Temporal compliance refers to the ability to cross in time. Spatial compliance is achieved by walking in a straight line. Velocity compliance refers to crossing at a regular walking pace. Visual search relates to head movements and attention direct toward traffic-related elements.

Table 2 presents the variables included within each of the compliance measures and their associated number of observations. With respect to temporal compliance, it should be noted that, at traffic lights without pedestrian lights, we considered crossings ending on yellow lights as ‘out of time’.

### 3.4. Statistical Analyses

First, Chi-squared tests provided an overview of the factors related to each of the four rule compliance measures. Relationships were further explored through four mixed-effect logit models, one for each compliance rule, controlling for age group and sex. Since many observations are recorded at each of the crosswalks, mixed-effect regressions enabled us to account for the grouping of observations in crosswalks using a random effect. Multivariate analyses were performed using Stata 12 with the *melogit* command. Odds ratios were calculated and are shown in the table in Section 4.3. An odds ratio over one means the variable increased the odds of complying with the measure. We also evaluated the marginal effects (*p* < 0.1), which facilitate interpretation of results [61] and inform future research.

A few variables had to be removed from specific models because they were a direct component of the dependent variable and, thus, an obvious problem of endogeneity would arise. After verifying for multicollinearity with Crammer’s V and VIF > 5, we excluded two variables: gender of adult (correlated with supervision) and head movements towards vehicles (correlated with car interaction). Supervision was also recoded for the multivariate analysis into a binary variable indicating whether the child was physically close (contact or within reach) or not (out of reach or no supervision). The rest of the variables were added to all models, including age group and gender.

## 4. Results

More than 700 children (n = 731) were observed at the 17 crosswalks. For temporal compliance, only the observations recorded at intersections with traffic lights were used (n = 568). Between 70% and 80% of children complied with at least one indicator. However, only a third of the observed child pedestrians complied with all of the indicators, both for controlled and uncontrolled crosswalks.

### 4.1. Individual, Situational, and Behavioral Categories

Table 3 presents the descriptive statistics. Two individual characteristics were recorded for each pedestrian: age and gender. Age was estimated in two categories: younger (approximately less than 9 years old) represented 57% of our sample, and older children (~ 9 to 12 years old), the remaining 43%. Although we did not conduct systematic observations, our samples were almost equally divided between boys (51%) and girls (49%). We recorded the adult gender and the level of physical proximity for children accompanied by an adult—i.e.,: whether there was physical contact, and whether the child was within the adult’s reach. According to our observation, 84% of children were supervised by adults, and 13% of these children were out of the adult’s reach. It is worth noting that a female adult accompanied half of the children, while 14% of them were with both male and female adults. We also recorded the number of other pedestrians crossing at the same time as the child pedestrians (outside of their party). Based on our samples, 37% of children crossed the street at the same time as other pedestrians (outside of their party), with a few of them (8%) crossing with six other pedestrians or more. In the present study, we included only a binary variable to capture interactions—meaning whenever the child and vehicle’s paths would cross while the child was still on the crosswalk. This allowed us to broadly examine how a vehicle crossing a child’s path affects rule compliance. We observed that 82% of child pedestrians did not experience any interactions with vehicles while crossing the street.

For intersections with traffic lights, ‘Stopping at the curb before crossing’ indicates whether the child waited for the next green light. We found that 59% of children stopped at the curb; Moreover, only 36% of them looked towards the vehicles, while more than half of them (53%) looked straight ahead/at the traffic light before starting to cross. We also recorded the ‘initiator of the crossing’ referring to the pedestrian, adult or child, who led the crossing. When there was no obvious initiator, the observer selected ‘adult and child at the same time’. In 57% of our observations, the adult and the child started to cross at the same time, and in 34% of cases, the adult was the initiator.

### 4.2. Univariate Analysis

Table 3 presents the results of univariate analysis for the rule compliance indicators and for each individual, situational, behavioral and road environment characteristic. Although younger children crossed in a straight line (spatial compliance) more often than older ones, they gave less visual attention to road-related elements (visual search). There was no statistical difference between boys and girls.

Where situational characteristics are concerned, the majority of children (84%) were accompanied by adults; among them, 71% were holding the adult’s hand or were within reach. The outcomes indicated that the presence of an adult impact the children’s type of crossing (i.e., spatial compliance), and less children were observed to follow visual compliance when an adult was present. The adult’s gender only seems to have an impact on visual search, since the children accompanied by female adults were more likely to look at road-related elements. More children were observed to follow temporal and velocity compliance when other pedestrians were crossing at the same time, but having groups of six or more pedestrians decrease the number of children following spatial and visual compliance.

If there is a car interaction, the child is less likely to cross in time (temporal compliance) and less likely to adopt a regular pace throughout the crossing (velocity compliance). However, he/she is more likely to comply with the visual search and spatial compliance indicators.

With respect to behavioral factors, 59% of children stopping at the curb before crossing associated with all rule compliance measures except visual search (not significant). Children who looked straight ahead or at the traffic light before crossing were more likely to demonstrate spatial compliance.

Almost half of children crossing with an adult did not have a noticeable initiator while 40% of crossings were initiated by the adult and 12% by the child. When a child initiated the crossing, he or she was less likely to comply with velocity compliance. When an adult initiated the crossing, the child was less likely to comply with temporal compliance and visual search.

The presence of a pedestrian countdown display (47% of crosswalks and 32% of crossings) was almost always associated with more rule compliance while the absence of signage is associated with less rule compliance. An outstanding 93% of children who crossed at an intersection with a pedestrian countdown display finished crossing in time. This proportion dropped to 70% for children who crossed at a traffic light without a pedestrian signal. A child crossing a street with a pedestrian light was less likely to walk in a straight line than a child crossing a street with only a traffic light. Crosswalks of mid-sized width associated with more spatial compliance and visual search, and negatively related to temporal compliance. Higher required speed to cross in time is negatively associated with temporal compliance: 13% of children did not finish crossing in time at crosswalks with speeds under 1 m/s, while this proportion rises to 46% at crosswalks with speeds over 1 m/s. Finally, more children were observed to follow all rule compliances when the distance between the nearest entrance of the park and the crossing was greater.

### 4.3. Mixed-Effects Logistic Models

To account for the clustering of observations by crosswalk, our binary measures of rule compliance (yes/no) were modeled in four different mixed-effect logistic regressions (temporal, spatial, velocity and visual compliance), controlling for age group and gender (see Table 4).

For *temporal compliance*, not many individual and situational variables were significant except for the car interaction, which decreases the odds of crossing on time. Out of all the variables, stopping at the curb (waiting for the next green light) has the highest odds of being associated with crossing in time (temporal compliance). The presence of a pedestrian countdown display also increases the odds of finishing in time by 3.6. However, an adult initiating the crossing decreases the odds by more than 40%. As expected, a higher required speed to cross on time is negatively associated with temporal compliance: a speed of more than 1 m/s reduces the odds of finishing in time by 70%.

As for *spatial compliance*, the physical presence of an adult and the interaction with a car increases the odds of crossing in a straight line; however, having big groups of pedestrians crossing at the same time (i.e.,: six or more) reduces the odds of complying with the measure. Spatial compliance shows increased odds with behavior like stopping at the curb before crossing and looking at traffic and the light before crossing. A medium-sized crosswalk (between 15 and 24 m) and a traffic-light controlled intersection also increase the odds of complying spatially.

With regard to *velocity compliance*, older and supervised children have higher odds of keeping a constant speed throughout the crossing. Using crosswalks with traffic lights and, when done, stopping at the curb before crossing, also increase these odds. However, the odds of keeping a constant speed are 65% less for crossing initiated by a child.

Older children had higher odds of *visual compliance*. Crossings involving car interactions, crosswalks of mid-sized width or intersections with stop signs or pedestrian countdown displays also increased the odds of *visual compliance*. When the child initiated the crossing, his/her odds of looking at road-related elements increased by 1.8 whereas when adults initiated crossing, the child’s odds decreased by half.

## 5. Discussion

### 5.1. Crossing Characteristics: Many Significant Factors

Several road elements have significant associations with the four indicators of rule compliance. As expected, children were more likely to finish on time at shorter crosswalks. When it came to wider crosswalks, they were more likely to conduct better visual searches and walk in a straight line (also significant at crossings with traffic lights), which echoes previous research findings indicating that children are more conservative in their behaviors when they are exposed to faster and denser traffic [42,47,62]. However, it should be considered that children are significantly more at risk in high traffic areas than low traffic districts [27].

Since the Manual of Uniform Traffic Control Devices for Canada (MUTCD) [63] and other similar manuals recommend 1.2 m/s as the recommended speed for crossing a street where traffic signals are present, it was no surprise that children were less likely to meet temporal compliance at signalized crosswalks where the required speed was over 1 m/s. Indeed, many scholars believe that a crossing speed of 1.2 m/s is too fast for most pedestrians [48]. In this respect, Deluka-Tibljas et al. [64] recommended that the design speed for signalized crosswalks near park for children under 11 years old should be 0.9 m/s and the length of the crosswalks should not exceed 7.0 m.

Our results show that higher levels of signage such as pedestrian countdowns are generally associated with increased rule compliance, which is in line with previous research findings [65]. Countdown displays seem to have a considerable impact on temporal compliance: when informed of the time left to cross in time, pedestrians may accelerate their walking speed accordingly to finish on time [52,55]. At intersections with traffic lights, children were more likely to walk in a straight line, which reinforces the idea that when exposed to heavier traffic, children adopt behaviors that are considered more careful.

### 5.2. Children’s Characteristics and Behaviors: Age Group, Stopping at the Curb and Head Movements

Age group was found to be the only significant individual factor in child pedestrian rule compliance. Even with potential recording of the observed pedestrian in the wrong group, the older children tended to show a more effective visual search and a more constant walking pace, which is like other research results [23,66,67]. Previous studies indicated that only up to the age of 12 or 13 do children’s road-crossing performance behaviors improve; once they pass this milestone, they are cognitively on par with adults but even then, lack of rule compliance has been observed [68].

Neither the gender of the child, nor that of the accompanying adult had a significant impact. These findings are similar to Wang et al. [28] the analysis of the full sample of children grade 1 to 6. However, when it was analyzed based on different age groups, boys and girls in the middle grades (3–4) had several significant gender-based differences.

Children who stop at the curb have more time and make better and more reasonable crossing decisions. These results are consistent with others who have found that stopping at the curb and waiting for the next green light before crossing increases the odds of crossing in time and allows the pedestrian to walk at a constant speed without having to rush [69]. Looking at road-related elements prior to crossing is also in keeping with the results of previous studies: pedestrians who are visually aware are more likely to comply with the rules [70,71].

### 5.3. Adult Supervision and Car Interaction

Where situational factors are concerned, our results are consistent with previous research studies on supervision: children who are physically close to adults are more likely to maintain a regular pace [44] and walk in a straight line [31]. These findings are reasonable considering that the physical supervision of an adult creates an inhibitory control on a child’s behaviors. To the best of our knowledge, no previous research study examines the impact of the crossing initiator—adult or child—on rule compliance. Our results provide evidence that whenever adults initiate crossing, children are less likely to pay attention to road-related elements and are less likely to finish crossing on time. Children who are supervised, as opposed to those who are alone, may sometimes display careless behaviors because they rely on adults for their safety [31,44,72]. Likewise, whenever children initiate crossing, they are more likely to perform a visual search because they are responsible for their own safety in this situation. Seemingly, children who initiate crossing are also more likely to change their walking pace. It can be hypothesized from our field observations that these children, already excited about going to the park, initiate crossing and also accelerate while crossing in order to reach the park faster.

As shown in other research [73], car interaction and red-light violation are directly associated: we found that car interaction decreases the chance of crossing in time, which might be due to children changing their behavior in order to avoid or manage interactions with the vehicles. As such, children who experienced a conflict with approaching vehicles considerably increased their visual search. Indeed, pedestrian–vehicle conflict risk can be compensated by an appropriate visual search from both the pedestrian and the driver [73]. Along the same lines, we found that interactions increase the odds of spatial compliance which may also be explained as a compensatory safe behavior from children since the proper usage of (marked) crosswalks can reduce interaction with vehicles [45]. Finally, we found that children were more likely to change their walking pace when they experienced a traffic interaction, which has also been reported by Pasanen and Salmivaara [74].

The main implication of our results about interactions has to do with risk perception: it is long established that risk perceptions are shaped by many factors, including our past experience [75,76,77]. A fifth of the children we observed had an interaction with a car that might have changed their or their parent’s crossing and walking experience. This kind of event might have implications such as less walking and more driving, implications that were not measures in the present project, but that are worth mentioning if we want to encourage active transportation in urban settings.

### 5.4. The Park as a Destination: Does It Influence crossing Behavior?

Although urban parks are undeniably popular destinations for children, the scientific community has paid very little attention to them when studying pedestrian road safety. None of the variables directly related to parks were significant in our models, but the presence of a park nearby seems to have a singular impact on the behaviors of child pedestrians. During our observations, parks had stimulating, yet less predictable, effects on the crossing behaviors of child pedestrians, including sudden acceleration, and more agitated head movements. For example, out of the 28% of children who changed their walking speed (tempo) while crossing, the vast majority was accelerating (84%) toward the park. Moreover, out of the 17% of children who were running in the park after crossing, 75% were already running beforehand, right in the middle of the street. Granié [31] found the opposite when studying child pedestrians near schools: 68% of them did not run while crossing toward schools.

### 5.5. Implications for Policy and Practice

This study explores the crossing behaviors of child pedestrians on roads around parks through an observational survey of individual, situational, behavioral and road environment predictors of pedestrian rule compliance. We already know that conflicts and interactions are, to some extent, related to collisions [40] and changes in the built and road environment reduce the risk of collisions [78]. Reducing the length of the crosswalk and increasing the time allowed to cross at signalized intersection by reducing the “targeted” walking speed are two examples of measures highlighted by our results that can be taken at the local level to improve the pedestrian experience when crossing. The findings of the study suggested that interaction with cars can impact children walking experience to go to parks. Urban planners should be aware of the volume and speed of cars around parks if cities want to encourage active living and active transportation towards such facilities. In addition, we found that presence of countdown and traffic lights can help children to cross the roads safely. To increase children’s safety and promoting walking to go to public spaces, neighborhoods parks need to be surrounded by better-designed crosswalks taking into account their physical and behavioral characteristics.

### 5.6. Study Limitations and Strengths

Although our results are informative and relevant to child pedestrian injury prevention, they have four limitations. First, as with any field survey, limitations from the data gathered through observations exist. For example, there is a possibility that the observers misreported age, even though few categories were used. In order to minimize this point, observers were trained onsite for several hours, but children can be much smaller or larger than average, which means that the older age group might represent not just older children but also taller ones. Second, for many predictors, any assumption of causality would be erroneous. For instance, whenever a car interaction arose, did it make the child more visually aware, or was the car interaction just minor collateral of what would have otherwise been a more severe conflict had it not been for the visual awareness of the child? Third, to what extent are the child pedestrians who comply with road rules safer? Adult pedestrians at fault have been associated with more severe injuries [79,80], but we did not find any conclusive results demonstrating that children who comply with pedestrian rules are safer since it was not the objective of our work here. The objective here was to examine individual, situational, behavioral and road environment characteristics relations to compliance with various road safety rules during street crossings. It can be seen as a first step before examining the actual relation between rule compliance and safety. This issue can be addressed in future research by considering all road users in a single framework (adults and children rule compliance: is there a difference?) and by focusing on pedestrian–vehicle conflicts (are pedestrians complying less involved in interactions?). Fourth, the growing number of potential distractions from the road environment and using technology (smartphones, etc.) has been indicated in recent works [65,81], but was not considered in the present study. This is another behavior that could be part of our observation tools.

What distinguishes this study from others is that it fills the gap regarding child pedestrian safety around parks by examining individual, situational, behavioral and road environment characteristics in relation to compliance with various road safety rules during street crossings. Despite the very limited literature on child pedestrian rule compliance, let alone child pedestrian rule compliance around parks, past studies that focused on adult safety at street intersections allowed us to create an analytical framework to fill this gap. We already know that education programs targeting children have little, short-term impact on pedestrian behavior or knowledge [21,40,78,82]; therefore, our results on non-individual characteristics influencing rule compliance are of interest. In addition, this study is one of a few studies that address the prevalence of traffic violations among pedestrians based on age and gender. However, we found that gender had no significant impact, but older children tended to show a more effective visual search and a more constant walking pace.

## Figures and Tables

**Figure 1 ijerph-19-13784-f001:**
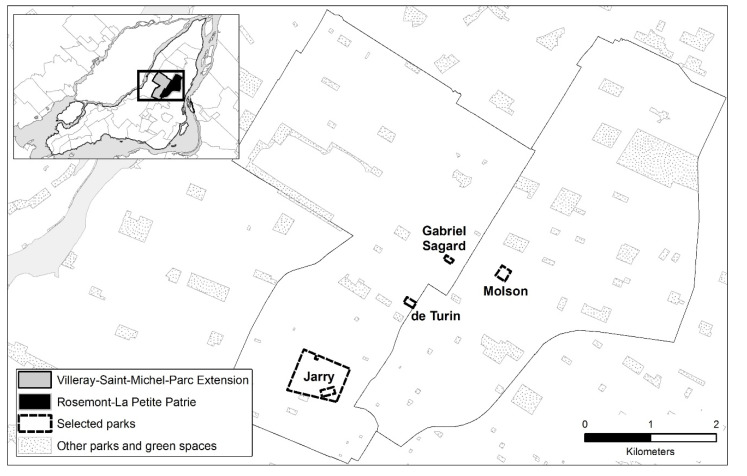
Location of selected parks on the Island of Montreal.

**Table 2 ijerph-19-13784-t002:** Rule compliance indicators.

	Compliance	Non-Compliance
Temporal	Crossing finished on Green light, white man or flashing red hand	Crossing finished on Red light, yellow light or red hand
Spatial	Type of crossing Crossed in a straight line	Type of crossing Crossed outside the parallel lines or diagonally
Velocity	Tempo Regular pace throughout crossing	Tempo Non-regular pace before or during crossing
Visual search	Head movements Head towards the traffic light, straight ahead or towards the vehicles before crossing	Head movements Head towards the ground, towards other pedestrians, towards an object or towards nothing in particular before crossing

**Table 3 ijerph-19-13784-t003:** Descriptive statistics and univariate analysis (χ^2^) for the 4 rule compliance variables.

	Total	Temporal		Spatial		Velocity		Visual	
	N (%)	Compliance N (%)	*p*	Compliance N (%)	*p*	Compliance N (%)	*p*	Compliance N (%)	χ^2^
Total		451 (79.4%)		541 (74.0%)		527 (72.1%)		512 (70.0%)	
Individual characteristics									
Age			0.77		0.002		0.877		0.001
Younger children (~4 to 8 years old)	416 (56.9%)	265 (58.7%)		326 (60.3%)		298 (56.6%)		272 (53.1%)	
Older children (~9 to 12 years old)	315 (43.1%)	186 (41.2%)		215 (39.7%)		224 (42.4%)		240 (46.9%)	
Gender			0.608		0.138		0.420		0.513
Girl	357 (48.8%)	224 (49.6%)		273 (50.5%)		250 (47.4%)		246 (48%)	
Boy	374 (51.2%)	227 (50.3%)		268 (49.5%)		272 (51.6%)		266 (52%)	
Situational characteristics									
Supervision			0.938		0.001		0.001		0.014
No adult	117 (16.0%)	51 (11.3%)		69 (12.7%)		78 (14.8%)		95 (18.5%)	
Adult but out of reach	94 (12.9%)	59 (13%)		70 (13%)		35 (6.7%)		66 (12.9%)	
Adult within reach or contact	520 (71.1%)	341(75.6%)		402 (74%)		409 (77.6%)		351 (68.5%)	
Gender of accompanied adult			0.110		0.200		0.400		0.030
Male	144 (19.7%)	103 (22.8%)		104 (19%)		106 (20.1%)		94 (18.3%)	
Female	366 (50.1%)	220 (48.8%)		290 (53.6%)		258 (49%)		262 (51.2%)	
Both	104 (14.2%)	77 (17%)		78 (14.4%)		80 (15.1%)		61 (12%)	
Other pedestrians			0.001		0.001		0.009		0.330
No other pedestrians	457 (62.5%)	229 (50.7%)		350 (64.7%)		313 (59.4%)		329 (64.2%)	
1 to 5 people	213 (29.1%)	166 (36.8%)		165 (30.5%)		156 (29.6%)		142 (27.7%)	
6 people or more	61 (8.3%)	56 (12.4%)		26 (4.8%)		53 (10%)		41 (8%)	
Car interaction			0.001		0.079		0.076		0.001
Yes	135 (18.5%)	70 (15.5%)		108 (20%)		88 (16.7%)		110 (21.4%)	
No	596 (81.5%)	381(84.5%)		433 (80%)		434 (82.3%)		402 (78.5%)	
Behavior characteristics									
Stopping at the curb before crossing			0.001		0.014		0.006		0.557
Yes	432 (59.1%)	319 (70.7%)		334 (61.7%)		325 (61.63%)		299 (58.4%)	
No	299 (40.9%)	132 (29.2%)		207 (38.3%)		197 (37.4%)		213 (41.6%)	
Looked straight ahead/at traffic light before crossing			0.722		0.002		0.256		
Yes	385 (52.7%)	250 (55.4%)		303 (56%)		268 (50.8%)		-	
No	346 (47.3%)	201 (44.6%)		238 (44%)		254 (48.2%)		-	
Looked towards the vehicles before crossing			0.127		0.499		0.001		
Yes	266 (36.4%)	140 (31%)		193 (35.7%)		170 (32.2%)		-	
No	465 (63.6%)	311 (69%)		348 (64.3%)		352 (66.8%)		-	
Initiator of the crossing			0.019		0.135		0.001		0.001
Adult and child at the same time	413 (56.5%)	253 (56%)		296 (54.7%)		294 (55.8%)		311 (60.7%)	
Child initiator	73 (10.0%)	48 (10.6%)		60 (11%)		35 (6.6%)		61 (11.9%)	
Adult initiator	245 (33.5%)	150 (33.2%)		185 (34.2%)		193 (36.6%)		140 (27.3%)	
Physical environment characteristics									
Signage			0.001		0.001		0.002		0.06
No signage	81 (11.1%)	-		56 (10.3%)		48 (9.1%)		52 (10.1%)	
Stop sign	82 (11.2%)	-		53 (9.8%)		50 (9.5%)		65 (12.7%)	
Traffic light without pedestrian light	332 (45.4%)	232 (51.4%)		282 (52.1%)		241 (45.7%)		222 (43.3%)	
Traffic light with pedestrian countdown display	236 (32.3%)	219 (48.5%)		150 (27.7%)		183 (34.7%)		173 (33.8%)	
Crosswalk width			0.046		0.001		0.240		0.077
Less than 15 m	296 (40.5%)	147 (32.6%)		192 (35.5%)		207 (39.3%)		195 (38%)	
Between 15 and 25 m	353 (48.3%)	229 (50.7%)		303 (56%)		250 (47.4%)		261 (51%)	
More than 25 m	82 (11.2%)	75 (16.6%)		46 (8.5%)		65 (12.3%)		56 (11%)	
Required speed to cross in time			0.001						
1 m/s or less	432 (59.1%)	378 (83.8%)		-		-		-	
More than 1 m/s	136 (18.6%)	73 (16.1%)		-		-		-	
Distance between the nearest entrance and intersection			0.150		0.001		0.016		0.141
5 m or less	317 (43.4%)	213 (47.2%)		214 (39.5%)		241 (45.7%)		213 (41.6%)	
More than 5 m	414 (56.6%)	238 (52.7%)		327 (60.5%)		281 (53.3%)		299 (58.4%)	

**Table 4 ijerph-19-13784-t004:** Mixed-effects logistic models of rule compliance (Odds ratios).

	Temporal	Spatial	Velocity	Visual
Age				
Younger [Ref.]				
Older	0.964	0.765	1.581 **	1.465 **
Gender				
Girl [Ref.]				
Boy	0.779	0.913	1.150	0.964
Supervision				
No [Ref.]				
Yes	0.901	1.817 ***	3.305 ***	1.017
Other pedestrians				
Alone [Ref.]				
1–5 people	1.383	1.083	0.838	0.807
6 people or more	1.830	0.434 **	1.285	0.855
Car interaction				
No [Ref.]				
Yes	0.468 ***	1.657 *	0.560 **	2.370 ***
Stopping at the curb before crossing				
No [Ref.]				
Yes	3.796 ***	1.458 *	1.456 *	0.754
Looks at the traffic light/straight ahead				
No [Ref.]				
Yes	0.862	1.562 **	0.829	-
Initiator of the crossing				
None [Ref.]				
Child	0.731	1.725	0.356 ***	1.789 *
Adult	0.526 **	0.960	1.186	0.469 ***
Crosswalk width				
Less than 15 m [Ref.]				
Between 15 m and 25 m	-	2.307 ***	0.887	1.88 **
More than 25 m	-	0.947	0.978	1.03
Signage				
No signage [Ref.]				
Stop sign	-	1.658	1.569	2.186 *
Traffic light without pedestrian light	-	2.080 *	2.003 **	1.119
Traffic light with pedestrian countdown display	3.577 ***	0.840	1.924 *	2.376 **
Speed required to cross in time				
1 m/s or less [Ref.]				
More than 1 m/s	0.301 ***	-	-	-
Distance between nearest entrance and intersection				
5 m or less				
More than 5 m	1.813	1.010	0.712	1.490 *
Constant	8.910 ***	0.745	0.704	1.124
Crossing site constant	0.000	0.071	0.000	0.000
Number of groups	13	17	17	17
Number of observations	568	731	731	731
Chi square	89.81	71.59	77.63	64.86
AIC	477.346	762.471	819.083	851.139

* *p* < 0.1,** *p* < 0.05, *** *p* < 0.01.

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
