# Peer review of "An Examination of Child Pedestrian Rule Compliance at Crosswalks around Parks in Montreal, Canada"

_ijerph, 2022, doi:10.3390/ijerph192113784_

Round 1
Reviewer 1 Report
This study explores the crossing behaviors of child pedestrians on roads around parks through an observational survey of individual, situational, behavioral and road environment predictors of pedestrian rule compliance. This study has provided detailed and robust field work and also has many confirmative results. However,my biggest concern in this paper is in regards to the contributions. It might be better if you can specify what we have known in this area and what you have done (research novelty) in this research. This paper created four binary composite indicators that distinguished child pedestrians and analyzed the factors associated with rule compliance. You may need to clarify why did you make such division. Is it necessary and what is the suggestions for children/adult and policy makers?
Specific comments can be found as below.
The authors are very familiar with the literature and cite many important works and scholars on factors associated with child pedestrian safety and compliance.
In your method section, you have provided a very robust observation process. More explanations are needed to better understanding how did you came to the four rule compliance.
In your result section, my biggest concern is regard to your presence of your data.
1) In Table 3, what is % represented for in the four rule compliance variables? For example, 451 (79.4%) Temporal compliance in total. What is the 79.4% represented for?
2) In your Chi-square analysis you present the significance as well as the positive/negative relationships? How did you come to the positive/negative correlations? As far as I see, the data you presented in table 3 are difficult to support your analysis.
In the result section and discussion section, you would better specify the contributions and policy implications in the end of your study.
Author Response
We thank this reviewer for the insightful comments, below are our response:
Reviewer #1: This study explores the crossing behaviors of child pedestrians on roads around parks through an observational survey of individual, situational, behavioral and road environment predictors of pedestrian rule compliance. This study has provided detailed and robust field work and also has many confirmative results.
However, my biggest concern in this paper is in regards to the contributions. It might be better if you can specify what we have known in this area and what you have done (research novelty) in this research.This paper created four binary composite indicators that distinguished child pedestrians and analyzed the factors associated with rule compliance. You may need to clarify why did you make such division. Is it necessary and what is the suggestions for children/adult and policy makers?
In the result section and discussion section, you would better specify the contributions and policy implications in the end of your study.
Authors: We have added two new sections to explain in more details the contribution and the strength of the study. Please see ‘implication for policy and practice’ and ‘study limitation and strength’ sections (p. ? L 101-104).
Reviewer #1: The authors are very familiar with the literature and cite many important works and scholars on factors associated with child pedestrian safety and compliance. In your method section, you have provided a very robust observation process. More explanations are needed to better understanding how did you came to the four rule compliance.
Authors:
The four rule compliance indicators are original constructs based on the combination of the literature review on children’s risky behaviour and data availability within our observation tool. We added this sentence within section 3.3
Reviewer #1: In your result section, my biggest concern is regard to your presence of your data. In Table 3, what is % represented for in the four rule compliance variables? For example, 451 (79.4%) Temporal compliance in total. What is the 79.4% represented for?
Authors: For temporal compliance, only the observations recorded at intersections with traffic lights were used (n=568). Of 568 children who were observed at intersections with traffic lights, 451 (79.4%) were recorded to follow temporal compliance. But for the rest of compliances 731 children were observed.
Reviewer #1: In your Chi-square analysis you present the significance as well as the positive/negative relationships? How did you come to the positive/negative correlations? As far as I see, the data you presented in table 3 are difficult to support your analysis.
Authors: The report of data in the result section has been edited in the way to better represent the data.
Reviewer 2 Report
The manuscript writed in good English and clear structure. The factors associated with child pedestrian were observed and analyzed carefully. The results are meaningful for children walking safety.
The manuscript ijerph-1869746 is the best article I have ever reviewed of IJERPH. The English expression, structure, literature review and discussion of the article are very good.
One more question for the author, why choose temporal, spatial, velocity and visual search compliance as the compliance indicators?
Author Response
Reviewer #2: The manuscript writed in good English and clear structure. The factors associated with child pedestrian were observed and analyzed carefully. The results are meaningful for children walking safety. The manuscript ijerph-1869746 is the best article I have ever reviewed of IJERPH. The English expression, structure, literature review and discussion of the article are very good. One more question for the author, why choose temporal, spatial, velocity and visual search compliance as the compliance indicators?
Authors:
The four rule compliance indicators are original constructs based on the combination of the literature review on children’s risky behaviour and data availability within our observation tool. We added this sentence within section 3.3
Reviewer 3 Report
This is interesting research. I agree with the authors that there is not much research in this area (especially regarding children). I appreciate the large research sample (731 children). The methodology is well described. I recommend that the main findings of the research are at least briefly stated in the last chapter of the manuscript.
Author Response
Reviewer #3: This is interesting research. I agree with the authors that there is not much research in this area (especially regarding children). I appreciate the large research sample (731 children). The methodology is well described. I recommend that the main findings of the research are at least briefly stated in the last chapter of the manuscript.
Authors:
We have added two new sections to explain in more details the contribution and the strength of the study. Please see ‘implication for policy and practice’ and ‘study limitation and strength’ sections. In addition, the outcomes have been described in different sections in discussion section to better represent the findings of the study.